# The molecular origin and taxonomy of mucinous ovarian carcinoma

Dane Cheasley 🆔 et al.#

Mucinous ovarian carcinoma (MOC) is a unique subtype of ovarian cancer with an uncertain etiology, including whether it genuinely arises at the ovary or is metastatic disease from other organs. In addition, the molecular drivers of invasive progression, high-grade and metastatic disease are poorly defined. We perform genetic analysis of MOC across all histological grades, including benign and borderline mucinous ovarian tumors, and compare these to tumors from other potential extra-ovarian sites of origin. Here we show that MOC is distinct from tumors from other sites and supports a progressive model of evolution from borderline precursors to high-grade invasive MOC. Key drivers of progression identified are *TP53* mutation and copy number aberrations, including a notable amplicon on 9p13. High copy number aberration burden is associated with worse prognosis in MOC. Our data conclusively demonstrate that MOC arise from benign and borderline precursors at the ovary and are not extra-ovarian metastases.

---

The origin of mucinous ovarian carcinoma (MOC) has long been controversial. It is now recognized that in the past, many mucinous tumors involving the ovary were in fact misdiagnosed metastases from diverse extra-ovarian sites, such as the colon, stomach, pancreas and uterus[1]. After revisions to the diagnostic criteria, the rate of mucinous tumors appearing to arise from the ovary fell from ~10% to only 3–5% of all epithelial ovarian cancers[2]. Nonetheless, it remains contentious if even these cancers represent occult extra-ovarian metastases[3]. Accurate diagnosis of primary MOC remains challenging, with a metastatic tumor from the lower gastrointestinal tract the most common alternative[4]. Knowing primary *versus* metastatic status strongly influences therapy selection, since most international guidelines indicate that first line therapy should be based on the tissue of origin[5–7]. Ovarian platinum-based therapies have low response rates for MOC[8] and because of the morphological similarities with colorectal mucinous tumors, a colorectal treatment regimen was proposed[9]. The difficulties in diagnosis alternatively led to the suggestion that all mucinous tumors could be treated with similar targeted therapies, regardless of origin[10,11]. However, both approaches assume molecular similarities across mucinous tumors, which is currently unknown.

Our current understanding of tumorigenesis incorporates a model of progression whereby the acquisition of cancer hallmarks is driven by the step-wise accumulation of genetic events[12]. The cell of origin for MOC remains unknown, however, mucinous benign cystadenomas and mucinous borderline ovarian tumors (MBT), the latter characterized by proliferative and atypical epithelial cells lacking stromal invasion, have been suggested as putative precursor lesions to MOC. These benign and MBT tumors can be cured by surgery alone[13] suggesting that they are ovarian in origin and an occult primary tumor at a distant site is unlikely.

Under a tumor progression model, we expect that if benign and borderline lesions are precursors to MOC they will share common initiating genetic events, and that development of a more aggressive phenotype, such as metastasis or higher tumor grade, will be accompanied by additional genetic aberrations. In support of this, previous work has found that MOC share genetic events such as KRAS mutations with benign and borderline mucinous ovarian tumors[14–16]. However, the contamination of earlier molecular data with metastatic tumors masquerading as MOC and the low sample size of more contemporary studies mean that the genetic events that drive invasive progression and metastasis of primary MOC remain largely unknown. In particular, due to its rarity and high probability of confusion with metastatic tumors from other sites, it is unclear whether MOC of high grade can develop from low grade MOC or other precursors.

In this study we ask two key questions: are MOC distinct from extra-ovarian metastases and primary mucinous tumors from other sites? What are the relationships between benign, borderline, invasive low-grade and high-grade mucinous tumors? To address these questions comprehensive genetic analyses are performed in a large multicenter, multinational cohort of these rare tumors. We find that MOC is distinct from tumors from other sites and identify a progressive model of evolution from borderline precursors to high-grade invasive MOC.

## Results

**Genetic analysis of mucinous ovarian tumors and other tumor sites.** We amassed over 500 potential mucinous ovarian tumors, including putative precursors, and undertook extensive pathological and clinical review to define a cohort of 255 primary MOC (Supplementary Data 1). Cases were ascertained from participating tissue banks and hospital databases as ovarian tumors with mucinous histology, and were reviewed with current diagnostic criteria to exclude mixed mucinous and non-mucinous ovarian tumors, and ovarian metastases from non-ovarian primaries (see Methods and Supplementary Fig. 1 for details of exclusions). Cases where the tumor was deemed likely to be metastatic but without a known primary site were also excluded. Whole exome sequencing was performed on primary MOC ($n = 48$), mucinous benign cystadenomas ($n = 5$) and mucinous borderline ovarian tumors (MBT, $n = 9$), including 24 sequenced in a previous publication[17]. Whole genome sequencing (WGS) was also performed for a subset of primary high-grade MOC ($n = 5$). Recurrently mutated genes identified in these 53 MOC were further investigated using a targeted sequencing panel in a cohort comprising MBT ($n = 20$), MOC ($n = 134$) and extra-ovarian metastases ($n = 23$) (Supplementary Data 2). The most frequent genetic event in MOC was copy number loss or mutation in CDKN2A (76% of cases), followed by mutations in KRAS and TP53 (both 64%). Amplification of ERBB2 (26% of cases) and mutations in RNF43, BRAF, PIK3CA and ARID1A (8–12% of cases) were the next most frequent (Fig. 1a).

It has been argued that even with current diagnostic practices most MOC, in particular high-grade cases, are metastatic tumors from other tissue sites[3]. Therefore, we compared our mucinous ovarian tumor sequencing data with that from TCGA and other available exome sequencing data. This comparison showed that MOC were clearly genetically distinct from high-grade serous ovarian, endometrial, gastric and colorectal tumors, including mucinous colorectal carcinomas and appendiceal neoplasms (Fig. 1b). Pancreatic adenocarcinomas were the most genetically similar to MOC, sharing the common combination of CDKN2A inactivation, KRAS and TP53 mutations. However, other events distinguish the two tumor types, with primary MOC carrying ERBB2 amplifications and RNF43 mutations, whereas pancreatic tumors show frequent SMAD4 alterations. Making it further improbable that MOC derive from pancreatic precursors, MOC 5-year survival rates - 82% (Grade 1) and 42% (Grade 3) - are significantly better than for metastatic pancreatic carcinoma (~1%) or for known extra-ovarian metastatic tumors in our cohort (Supplementary Note 1, Supplementary Tables 1–4, Supplementary Figs. 2–5).

**Progression of MOC from precursor tumors: mutation analysis.** Since our genetic data indicate that MOC are unlikely to be metastases from extra-ovarian sites, but rather *bona fide* ovarian carcinomas, we next assessed the genetic relationship between MOC and putative benign and borderline ovarian precursors. If MOC evolve from these non-invasive lesions, we expect them to share initiating events (those present at allele frequencies indicating they are present in all tumor subclones). Notably, the majority of MOC carried at least one initiating genetic event that was also present in most MBT cases. For example, a mutation in KRAS, BRAF and/or CDKN2A was observed in 95% of grade 1 (83/87), 88.6% of grade 2 (70/79) and 83% of grade 3 (20/24) MOC and was also present in 95% of MBT (Supplementary Tables 5 and 6).

Under a model of progression, we expect to observe evidence of additional genetic events driving invasion, increasing tumor grade and metastasis. We first evaluated whether the point mutational profile drove invasive progression. TP53 was the only gene to show enrichment of mutations in MOC (64%) compared to MBT (18%) (Fisher's exact test, two-sided, $p < 0.0001$, OR 0.12, 95% CI 0.045–0.31, Supplementary Table 5). Globally, MOC did not have a significantly higher point mutation burden than observed in the non-invasive lesions, nor did the number of variants differ with grade (Fig. 1c). Mutation signature detection revealed that most MOC had an age-related signature (COSMIC Sig1[18]), either as the major (73%) or a minor (22%) component (Fig. 1d). This

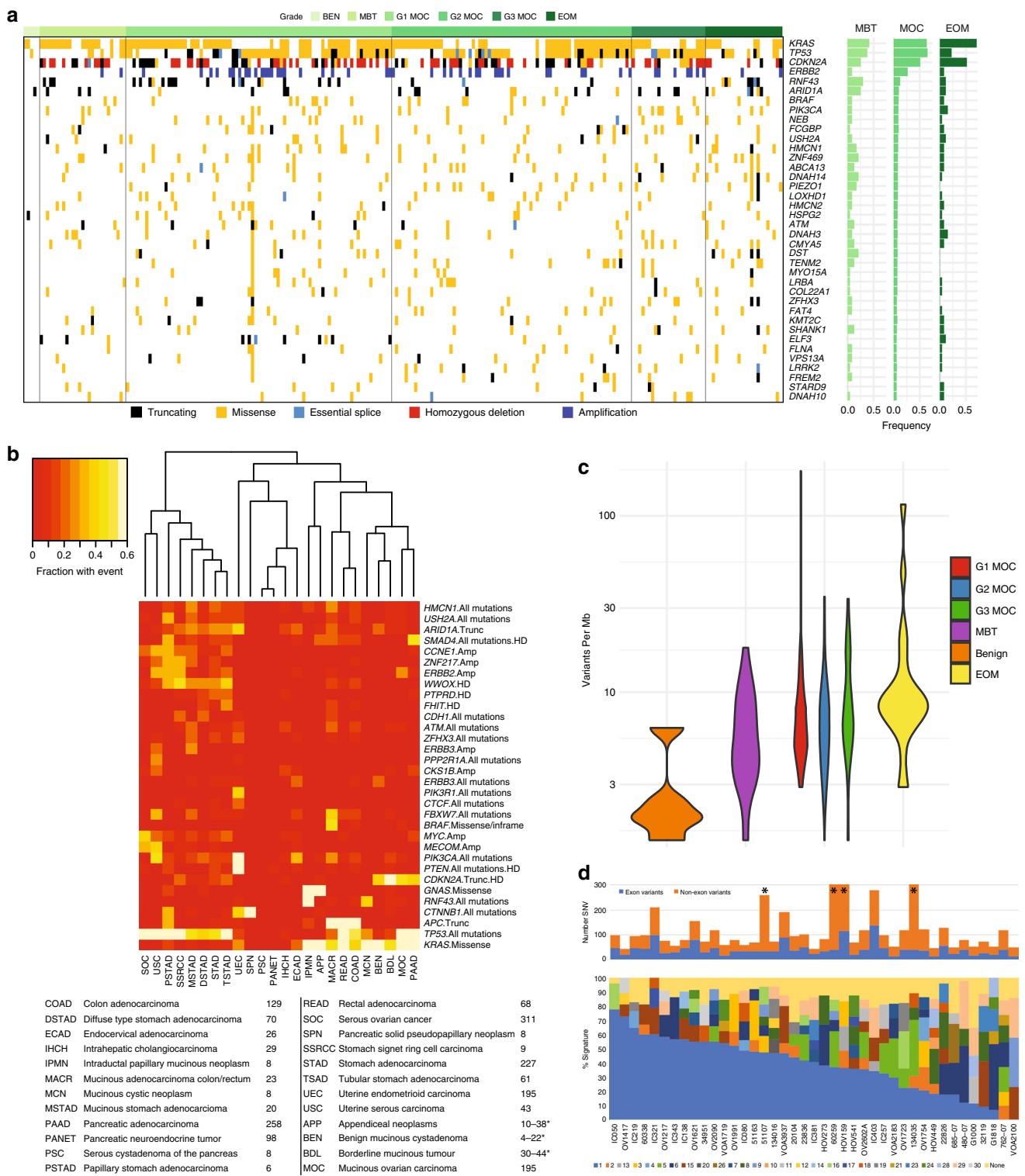

**Fig. 1** Variant analysis. **a** Summary of variants across the cohort for genes mutated in > 5% of MOC; also includes copy number alterations for *CDKN2A* and *ERBB2*. BEN, benign mucinous; MBT, borderline mucinous; MOC, mucinous ovarian carcinoma; EOM, extra-ovarian metastases. **b** Comparison of copy number alterations and mutations with other tumor types, summarised by frequency for each. Number of cases shown below. *Higher number is for selected genes tested by Sanger sequencing/SNP arrays (see Methods); lower number from exome analysis. **c** Number of variants per Mb by group (ANOVA, two-sided, F = 1.55, df = 5, p = 0.18), combining exome and targeted sequencing cohorts. **d** Top: number of single nucleotide variants (SNV) input to signature detection from whole exome and whole genome sequencing, with each column an MOC case. Note whole genome samples truncated at 300 (asterisk). Bottom: COSMIC mutation signatures[18].

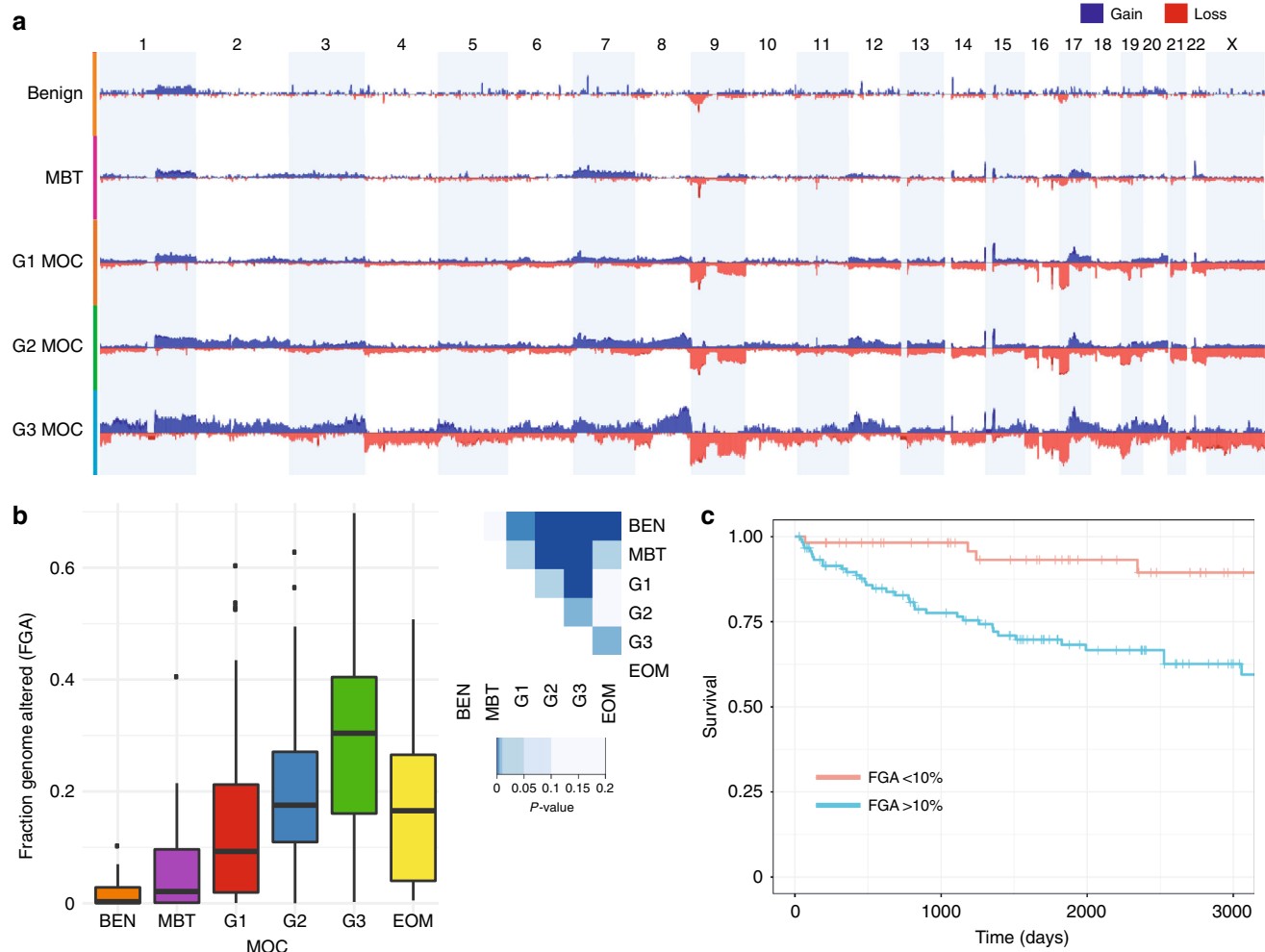

**Fig. 2** Copy number analysis. **a** Comparison of copy number frequency across the genome, comparing benign (BEN), borderline (MBT), MOC grade 1 (G1), grade 2 (G2) and grade 3 (G3). **b** Fraction of the genome altered (FGA) by group including extra-ovarian (EOM) mucinous tumors (ANOVA, two-sided $p <$ 0.001, F = 17.0, df = 5). Tukey post-test comparison $p$-values shown at right (two-sided). Error bars are "Tukey" (geom-boxplot in ggplot2). **c** MOC disease-specific survival by FGA. Score (logrank) test = 11.98 on 1 df, $p = 0.002$

spectrum was similar to benign tumors (100% Sig1 as the major component) and MBT (71% major, 29% minor Sig1). A similar signature pattern for MOC was also obtained through de novo mutation signature detection on five grade 3 MOC cases with WGS data. We found three identified signatures present in varying degrees in these cases – S1 (COSMIC Sig1; Age), S2 (COSMIC Sig2 and 13; APOBEC), and S3, resembling COSMIC Sig8 (Supplementary Fig. 6). This result indicates that MOC, including even high-grade MOC, share similar mutational profiles to benign and borderline precursors.

**Progression of MOC from precursors: chromosome aberrations**. As the point mutation spectrum was similar between MOC and the precursor lesions, we evaluated chromosome copy number aberrations as another genetic mechanism of progression. Data for over 250 cases (22 benign, 39 MBT and 195 MOC) show that MOC cases have more copy number alterations than pre-invasive disease (Fig. 2a). The fraction of the genome altered by copy number also increases significantly with grade (Fig. 2b), and is associated with patient outcome (Fig. 2c). Specific copy number alterations associated with progression from MBT to grade 1 MOC were losses at 9pter-p21.2 and 17p (Fig. 2a, Supplementary Dataset 3). Increased 17p loss is likely to reflect the increased prevalence of *TP53* mutations in MOC, as has been shown in other cancer types[19,20]. In

contrast, loss of the most obvious tumor suppressor gene on 9p, *CDKN2A* (9p21.3), is an early event in mucinous ovarian carcinogenesis, commonly seen in benign mucinous and MBT precursors. Indeed, loss of the *CDKN2A* locus was not itself significantly associated with progression (56.4% MBT, 69.7% MOC), but an expanded area of deletion encompassing most of the chromosomal arm was enriched in MOC. This result suggests that other tumor suppressor genes may be located on chromosome 9 that are important for invasive progression.

Interestingly, we observed that MOC with *CDKN2A* loss combined with *TP53* mutation were significantly more likely to have copy number amplification at 9p13.3 (Fisher's exact test, two-sided, $p < 0.0001$, OR 0.05, 95% CI 0.001–0.34, Fig. 3). This combination could be synergistic in driving MOC development, as the amplicon was not seen in MBT or benign tumors. A similar, but independent, association was observed between *TP53* mutation and *ERBB2* amplification, with this combination rarely observed in MBT (Supplementary Fig. 7). We evaluated the effect of 9p13.3 amplification on the transcriptome and found significantly increased expression of genes associated with chromosome condensation and kinetochores ($p = 4 \times 10^{-9}$, STRING Hypergeometric test, Supplementary Fig. 8). We speculate that 9p13.3 amplification could lead to enhanced genomic instability, as the fraction of the genome altered by copy number was higher in

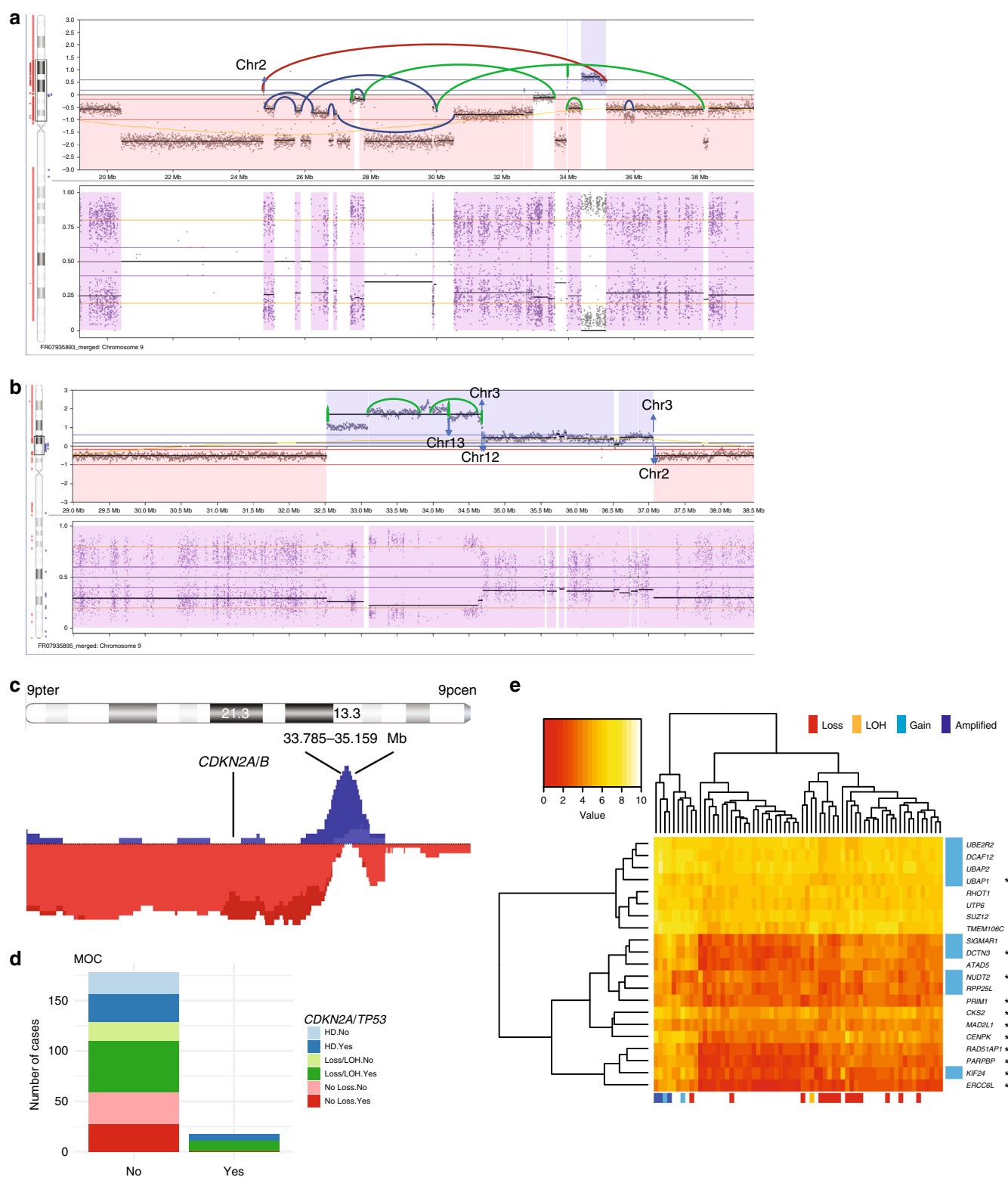

**Fig. 3** 9p13 amplification. **a** WGS case HOV159 showing complex amplification with the majority of breakpoints rejoining internally to chromosome 9. **b** WGS case 5950 showing simpler amplification with a mixture of internal and external breakpoint fusions. **c** Overview of all amplified samples (gains in blue, losses in red) showing minimal region of overlap at 33.785–35.159 Mb and association with 9p loss. **d** All MOC with 9p13 amplification have *TP53* mutation and 16/17 also have have *CDKN2A* loss/loss of heterozygosity (LOH). HD, homozygous deletion. For *TP53*: No: no mutation detected, Yes: mutation detected. **e** RNAseq analysis showing genes significantly differentially expressed between amplified and non-amplified (FDR < 0.05). Blue genes = within 9p amplicon, *genes connected in STRING network related to chromosome condensation

9p13.3-amplified samples than non-amplified samples. In contrast, cases with *ERBB2* amplification did not have a significantly increased copy number burden than *ERBB2* non-amplified cases (Supplementary Fig. 8, Supplementary Table 7).

Global copy number profiles fell into distinct types: simple (23% of MOC, compared to 82% of MBT), simple with one high-level amplification (9% MOC, 2.6% MBT), complex (23% MOC, 13% MBT), complex-whole chromosome (5% MOC, 2.6% MBT) and complex with multiple amplifications (40% MOC, 0% MBT). Grade 3 cases were significantly more likely than grade 1 to have a complex profile ($p = 0.03$, $\chi^2 = 14.1$, Chi-squared test; Supplementary Table 6), or amplifications on multiple chromosomes ($p = 0.03$ $\chi^2 = 10.5$, Chi-squared test). The mechanisms underlying these structural variants varied between cases as illustrated by four such grade 3 cases with WGS data (Fig. 4e, Supplementary Note 1, Supplementary Tables 8 and 9). For example, case 5950 (Fig. 4e, G3-B) had a majority of small intra-chromosomal aberrations (64.5% of breakpoints) including fold-back inversions (14%), whereas most structural changes in case 6987 (Fig. 4e, G3-C) were unbalanced rearrangements > 1 Mb in size (63.7%) including inter-chromosomal translocations (38.5%). There were no striking differences in specific regions between grade 1 and grade 2 cases or between grade 2 and grade 3, but many regions were markedly increased in frequency of aberration between grade 1 and grade 3, suggesting that grade 2 cases represent an intermediate molecular stage. The copy number alterations most strongly enriched in grade 3 MOC were gains of 1p and 19p, affecting multiple oncogenes including *JUN*, *JAK1*, *MYCL* and *BRD4* (Supplementary Data 3).

**Molecular evolution of Grade 3 MOC and metastatic disease.** We investigated the evolutionary process from MBT to high-grade metastatic MOC in a unique case of a patient who consented to the collection of her primary tumor tissue at initial diagnosis and, following recurrence with rapid deterioration in her condition, to the donation of multiple metastatic tissue samples ($n = 16$) collected via a rapid autopsy program (Fig. 4a, b)[21]. The patient was initially diagnosed with a Stage IA tumor, mostly MBT with small areas of invasive carcinoma (Fig. 4, Supplementary Note 1). She did not have any chemotherapy. At 26 months after diagnosis, she presented with widespread recurrence of infiltrative grade 3 disease and died shortly afterwards. WGS was performed on two histologically-distinct areas of the primary tumor (with MBT and grade 3 invasive morphology respectively) in addition to four regionally distinct metastatic sites (M1-4, Fig. 4b). Mutation analyses showed that the primary and recurrent tumors were clearly clonally related, sharing driver events including *TP53* and *KRAS* mutations, and *CDKN2A* homozygous deletion. However, genetic variants not observed in the primary tumor were found present across the metastatic sites with high concordance. Targeted sequencing for metastasis-specific variants indicated that 12/13 tested variants were present in at least 12 of the 15 metastatic sites assayed (Supplementary Note 1, Supplementary Table 10), suggesting that metastatic sites were all descended from a common ancestor that developed prior to rapid dissemination.

Copy number analysis showed similar concordance for chromosomal rearrangements among the four metastatic sites evaluated by WGS, but a dramatic difference was seen between the primary and recurrent tumors. The primary tumor had a relatively stable copy number profile, with 14% of the genome affected. Recurrent disease was associated with a near doubling of copy number alterations (24–26% fraction genome altered), and also a near doubling in overall ploidy (estimated at 1.9 for the primary tumor and 3.6–3.7 for the metastases). High-level amplifications were notably increased, from 1 to >10 events.

These amplification events encompassed 15 COSMIC oncogenes or fusion partners, including *MYC*, *ETV1*, *HMGA1*, and *CCND3*. While all structural variant types were increased in recurrent samples, large intra-chromosomal and fold-back inversions were present in greater proportions (Supplementary Note 1, Supplementary Tables 8 and 9). The increase in fold-back inversions from 4–6 (7–9%) in the primary tumor to 16–20 (12–13%) in the metastases is consistent with the increase in high-level amplifications. Thus, we propose that disease progression in MOC is characterized by an increase in copy number aberrations.

Consistent with the data from the rapid autopsy case, MOC diagnosed at Stage III or IV carried a higher fraction of the genome altered by copy number than Stage I carcinomas, even when grade was taken into account ($p = 0.007$, ANOVA with Tukey post-test, difference = 0.089, 95% CI = 0.02–0.16). Since copy number alterations were the most striking feature of metastatic progression, we evaluated whether they were related to patient outcome. In MOC, high structural genomic complexity (measured as copy number profile type (e.g. "simple", "complex" etc.) or the fraction of the genome altered) was significantly associated with worse overall disease-specific and progression-free survival (Fig. 2c, Supplementary Note 1, Supplementary Figs. 4 and 5). Copy number loss/LOH of 9p and amplification of 9p13 were also associated with poorer clinical outcomes, but *TP53* mutation, *KRAS* mutation or *ERBB2* amplification were not. In a multivariate model selection analysis[22], the fraction of the genome altered and 9p loss were significant factors, along with grade, *ERBB2* amplification and FIGO stage (Supplementary Note 1). The association of 9p loss with poorer clinical outcomes cannot be due to *CDKN2A* loss, as the latter is an initiating event in benign and MBT tumors where the outcomes are favorable. Therefore, other possibly haploinsufficient tumor suppressors on 9p as well as 9q may be responsible, as loss of these chromosome arms was more frequent in tumors with disease-specific mortality (Supplementary Fig. 9).

**Discussion**

We have generated a model of progression from benign to MBT to localized low-grade MOC and progressively through to high-grade and/or metastatic MOC. Benign tumors initiate with either a *KRAS* or *CDKN2A* event. MBT are significantly more likely to have both events and may have additional copy number alterations. Grade 1 MOC have yet more copy number alterations, and are more likely to have a *TP53* mutation. Copy number alterations are key drivers associated with increasing grade and metastatic progression, and are potential prognostic markers.

Our data showing that MBT precursors can beget high-grade MOC contrasts with high-grade serous ovarian carcinoma that hardly ever derive from borderline or low-grade disease. Indeed, the majority of grade 3 MOC had associated benign and/or borderline components (20/23, 86%), which is rare in high-grade serous ovarian carcinoma[23]. Data from epidemiological and genome-wide association studies support a common origin for MOC and MBT tumors, as the risk factors and SNPs are shared between the two conditions[24–26]. It is therefore crucial that this risk of progression is taken into consideration in the surgical management of women with MBT. It remains to be determined whether any of the genetic aberrations we have identified as important for progression could also be prognostic markers for MBT, for example *TP53* mutation or copy number burden.

Mutation in *KRAS* is a key early event for ovarian mucinous tumors. Oncogenic RAS has been linked to genomic instability through a number of mechanisms, including replication stress and shortened G2[27–29], but we did not see an association of *KRAS* mutations with genomic instability overall (Supplementary Table 7). However, survival of cells after events such as

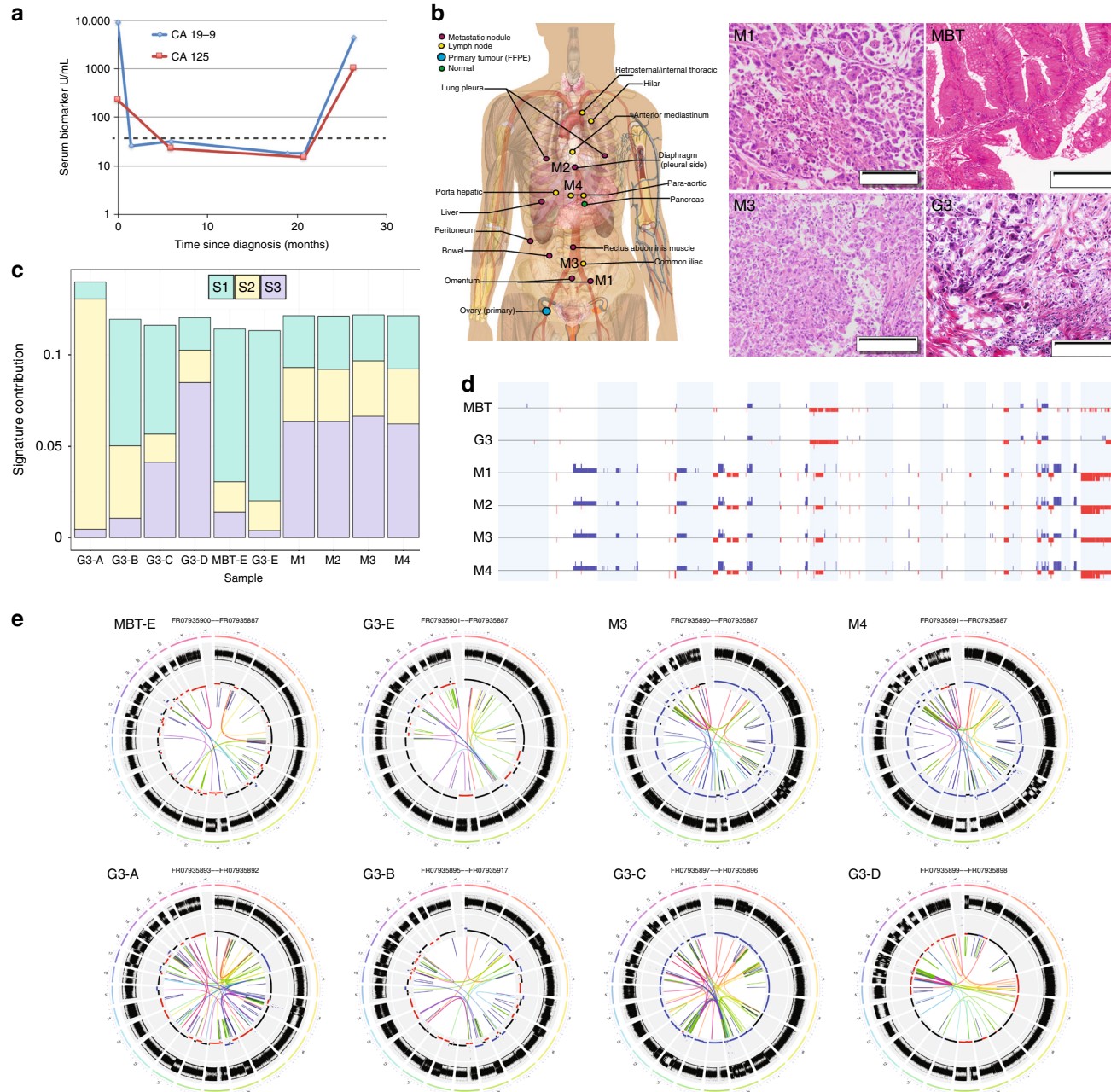

**Fig. 4** Molecular evolution of Grade 3 MOC and metastatic disease. **a** Serum markers of rapid autopsy case. Dashed line indicates maximum normal level. **b** Schematic of location of metastatic tissues taken at autopsy and haematoxylin and eosin stained sections of metastatic sites (M1 and M3) as well as two areas of the primary tumor, the majority borderline morphology (MBT) and the small area of high-grade invasive tumor identified in the frozen section (G3). Scale bars are 200 μm. **c** Mutation signatures (S1 - S3) identified by de novo analysis of 5 Grade 3 cases (A-E) showing the shift in mutation signature profile from primary (MBT-E, G3-E) to metastatic sites at autopsy (M1 - M4). **d** Copy number of primary and metastatic sites showing dramatic increase in structural and copy number alterations in the metastatic sites. Blue, gain; red, loss. **e** Circos plots illustrating structural variants in primary (MBT-E, G3-E) and metastatic sites (M1–M4) in the rapid autopsy case, and in four independent Grade 3 MOC cases (G3-A, G3-B, G3-C, G3-D). G3-A and G3-B have chromosome 9p amplicons

cytokinesis failure and genome tetraploidisation may require aberrant p53[30], and indeed *TP53* mutation was strongly associated with genomic instability measures in our cohort. The combination of *TP53* mutation and *KRAS* mutation in the rapid autopsy case may explain the survival of a tumor cell after a tetraploidisation event. This event may have enabled the increase in copy number amplifications in this case, as tetraploidisation has been shown to lead to increased structural variation from elevated replication stress[31]. Recently, a copy number signature that was associated with RAS pathway aberrations was identified

in high-grade serous ovarian cancer (in the context of near-ubiquitous *TP53* mutations)[32]. The presence of amplifications and fold-back inversions with relatively few other breakpoints in this signature is consistent with the type of copy number profiles observed in the rapid autopsy case as well as many other MOC, particularly those of high grade and/or *TP53* mutated.

In conclusion, our data do not support a non-gynecological origin for MOC as previously postulated[3], but it remains unclear whether the ovarian surface epithelium is the cell of origin for MOC. Other "ovarian" tumor histotypes arise from non-ovarian

cells, such as high-grade serous carcinoma (fallopian tube) and endometriosis-associated carcinomas (endometrioid and clear cell subtypes); MOC could be similar. This uncertainty suggests that a personalized molecular therapeutic approach could be more effective than a standardized protocol based on the tissue of origin. Indeed, our analysis identified several potential targets such as *RNF43* (Wnt-pathway inhibitors), *ERBB2* (HER2 targeted therapies), and potentially new dual RAS/RAF inhibitors targeting *KRAS*. Assessing the efficacy of targeted therapies specifically for this tumor type will be challenging given its rarity, such that basket trials assessing specific targets may be more practical and will be crucial to improve outcomes for women with advanced or recurrent MOC.

## Methods

**Cohort and pathology review**. We evaluated collections of mucinous tumors from 11 different sources in four countries (Supplementary Data 1): Australia: Royal Women's Hospital, the Victorian Cancer Biobank, The Hudson Institute of Medical Research (all Victoria); Garvan Institute, Westmead Gynaecological Biobank (New South Wales); Queensland Institute for Medical Research-Berghoffer (Queensland); Australian Ovarian Cancer Study (AOCS) and CASCADE (Australia-wide); United Kingdom: Edinburgh Cancer Research Centre; United States: the Mayo Clinic (MN); Canadian Ovarian Experimental Unified Resource (COEUR, Quebec, Canada[33]) and OVCARE (British Columbia, Canada). The 527 tumors assessed included 36 benign, 136 MBT, 318 invasive primary and 37 known extra-ovarian metastases (Supplementary Fig. 1).

The review process involved identifying frozen tissue with tumor ($n = 137$), and full pathology review when diagnostic slides were available. Cases that only had FFPE tissue available (used for targeted sequencing) underwent various levels of pathology review, mostly of 1–2 slides. Factors assessed were based on Lee and Young[34] and included: primary tumor size ( < or ≥10 cm), unilaterality vs bilaterality, presence of benign/borderline components, intact/smooth capsule/ surface involvement, any evidence of an extra-ovarian primary concurrently or historically, expansile vs infiltrative invasion, presence/absence of signet ring cells, microscopic cysts, complex papillae, necrotic debris, small glands/tubules, single cells, nodular growth, hilar involvement. Immunohistochemistry data was also reviewed if available. All these factors were considered collectively to reach a decision. Review was conducted by MC, JP, PEA, CBG, MK, RS and more detail on exclusions are provided in Supplementary Fig. 1. Grade was assessed considering the architectural pattern (glandular < papillary < solid), nuclear anaplasia (mild < moderate < severe), and mitoses/mm$^2$ (≤3, 4–7, ≥8).

**DNA and RNA extraction**. Tumor genomic DNA was isolated from cells obtained by needle-point microdissection of areas with >80% neoplastic cellularity from 10 μm hematoxylin and eosin (H&E)-stained tumor sections. For the discovery cohort (frozen tissues), DNA was extracted using the DNeasy Blood and Tissue Kit (Qiagen). Matched germline DNA was extracted from whole blood (36 cases) or uninvolved ovarian stroma (18 cases). For the validation cohort (FFPE tissues), microdissected tumor tissue was added to 180 μl Buffer ATL (Qiagen) with 20 μl proteinase K solution (>600 mAU/ml, Qiagen), and incubated at 56 °C overnight to completely lyse tissue. The following day tissue was incubated at 90 °C for 1 h to reverse formaldehyde modification of nucleic acids. Following incubation, 2 μl RNaseA (100 mg/ml, Qiagen) was added to sample and incubated for 2 min at room temperature. DNA Extraction and purification of DNA was then performed using the DNeasy Blood & Tissue Kit (Qiagen) spin-column procedure. For tumor RNA extraction, 10 μm sections were stained with 1% cresyl violet acetate followed by needle microdissection and RNA extraction using the miRNeasy Kit (Qiagen).

**Library construction and massively parallel sequencing**. Whole genome libraries (10 tumor, 5 germline) were constructed by the Ramiaciotti Centre for Genomics (UNSW, Sydney, Australia) from 500 ng of DNA with the Illumina TruSeq DNA Sample Preparation protocol (Illumina, San Diego, CA, USA). Each resulting paired-end library was sequenced on an Illumina HiSeqX. Whole exome libraries were constructed from 200 to 500 ng of DNA. Exome capturing was performed using the SureSelect Human All Exon kit V6 (Agilent Technologies, Santa Clara, CA) and massively parallel sequencing performed using the Illumina Hi-Seq2000 with 150 bp reads. Targeted sequencing of FFPE tumor DNA was performed using a custom SureSelect XT Custom Panel (Agilent, Supplementary Data 2). This panel targets 462 genes found to be recurrently mutated in our discovery exome analysis, including an additional 7 actionable genes, 25 genes associated with EOM sites, three MOC GWAS SNPs, one LGSC associated gene, and two additional RAS/RAF driver genes. Library preparation was performed using the KAPA Hyper Prep Kit (Kapa Biosystems). Sequencing of target-enriched DNA libraries was performed using the Illumina Next Seq 500 generating 75 bp paired-end sequence reads. Sequencing QC measures are provided in Supplementary Data 4.

In total 200 ng of RNA was used to generate libraries using the TruSeq Stranded Total RNA LT Sample Prep Kit with Ribo-Zero Gold (Illumina). Libraries were

pooled and single-end sequenced to 50 bp on an Illumina Hi-Seq2000 to achieve a minimum of $30 \times 10^6$ reads per sample.

**Mutation analysis**. Whole-genome sequencing data was processed using Seqliner WGS pipeline (v0.4; seqliner.org). Reads were aligned to GRCh37/hg19 using BWA-MEM (v0.7.10). Picard (v1.119) was employed to sort and index the alignment BAM files, and to mark duplicate reads. Genome Analysis Toolkit (GATK; v3.2) performed local realignment around indels and was used to recalibrate base quality scores. Somatic variants were called from whole genomes using VarDict (v1.4.6) and MuTect2 (v3.5). Germline variants were called using GATK HaplotypeCaller (v3.2).

Exome sequence variants were called using matched normal DNA when available, with a pipeline that included subtracting variants found in the normal DNA. Variants were called using GATK UnifiedGenotyper[35], Platypus[36] and VarScan[37]. Called variants were annotated using the Ensembl Variant Effect Predictor Release 78[38].

Variants were filtered to identify high confidence somatic variants as follows: 1) excluded if present in ExAc (minus TCGA samples), ExAc non-Finnish European or EVS at an allele frequency of ≤0.0001, unless listed in ClinVar as pathogenic or potentially pathogenic. The QUAL score had to be ≥30, read depth ≥ 10, alternative base read depth ≥ 2, and allele frequency of >0.05. For samples with a matching normal sample, the frequency of the alternative allele had to be ≤0.05 in the normal DNA. Variants were excluded if the gene was blacklisted in Scheinen et al.[39]; if the variant was present in more than one of our in-house collection of germline exomes ( >300 cases); or if present in >20% of the cohort except for known hotspot mutations (e.g. *KRAS* codon 12). These measures helped to reduce common technical artefacts. Variants were excluded if called only by VarScan, and for unpaired samples if also only called by UnifiedGenotyper. Variants also had to pass default caller filters. For targeted sequencing analysis, all cases lacked a matching normal, so the same filters were applied as for unpaired exomes, except the read depth had to be ≥20 and the variant had to be supported by ≥10 reads. In addition, variant had to be absent from three normal samples run on the same panel. Filtered variants are in Supplementary Data 5.

Tumors with exome data lacking a normal control DNA had more variants and were excluded from statistical analyses of mutation number and spectrum. In performing analyses comparing the number of variants from tumors of various types, only filtered variants in the validation panel genes were selected from exome and WGS data to minimize the effect of the sequencing platform. The number of variants was calculated using all coding sequence and splice-site variants and were converted to variants per Mb by dividing by the amount of target sequence in the capture (2.07 Mb). An ANOVA was performed using either Grade or Classification and sequencing type (exome paired, exome unpaired, WGS, validation) as factors. Tukey multiple testing correction was performed. Statistical analysis was performed in R (v3.3.0) and all tests were two-sided.

**Signature detection**. DeconstructSigs (v1.8.0) was performed on variants filtered as above. For de novo discovery of signatures on the WGS cases, variants were additionally filtered against Encode blacklist, RepeatMasker, presence in dbSNP, presence in Exome variant server (EVS). These high confidence variants were used in NMF signature discovery (R package SomaticSignatures[40]) with the number of signatures set to three.

**Copy number analysis**. Existing SNP array data was used when available. Copy number data was otherwise obtained from exome sequencing using ADTEx[41] using matching normal germline DNA as a baseline when available. For whole genome sequencing data, structural variations were predicted using MANTA (v1.0.3)[42] (Supplementary Data 3) and copy number aberrations and LOH were detected using FACETs (v0.5.6, cval 1500)[43]. For the targeted sequencing panel CopywriteR[44] was used for copy number with 50 kb bins, utilising a normal lymphocyte DNA control run in the same sequencing batch for the normalisation baseline (NA12878, Coriell Institute). Data was then imported into Nexus Copy Number$^{TM}$ (software v8.0, BioDiscovery Inc), segmented using a FASST2 segmentation algorithm and visualized (Supplementary Data 3). Thresholds were log$_2$ ratios of ±0.2 for gains and losses, >0.6 for high level gains and <−1 for homozygous deletions.

Comparisons between groups were performed using Nexus Copy Number using a p value threshold of <0.002 (selected by dividing a p value of 0.05 by 23 (number of chromosomes), since segments within chromosomes are not independent). A requirement for a percentage difference of at least 15% was also applied. Additional filtering was performed to exclude segments that were statistically significantly different between platforms i.e. to account for differential detection of artefactual segments by exome analysis compared to the targeted sequencing panel. Segments that overlapped with a copy number polymorphism (as determined by Nexus) by >90% were also excluded.

For classifying WGS structural variants, deletions and duplications were considered small if less than 1 Mb. Inversions were also considered small if <1 Mb, but called as fold-back inversions if <30 kb, following Wang et al.[45]. Deletions, duplications and inversions were called intra-chromosomal, all other translocations (denoted "BND" in MANTA annotation) were considered to be inter-chromosomal translocations.

Fraction of the genome altered was the number of bases affected by copy number change divided by the total size of each chromosome and then averaged across all chromosomes[46].

**Sanger sequencing**. Key variants identified by massively parallel sequencing were validated by Sanger sequencing. Normal DNA where available was also subjected to Sanger sequencing alongside matched tumor samples. Sanger sequencing primers (Supplementary Table 11) were designed using the Primer 3 tool[47] and target sequences amplified. The BigDye Terminator system (Applied Biosystems) was used for sequencing on a 3730 DNA Analyzer (Applied Biosystems). The sequencer output was viewed using Geneious 8.1.9 software (Biomatters, Auckland, New Zealand).

**Comparison to other tumor types**. Data were downloaded from cBio (April 2018) from provisional TCGA studies of ovarian, pancreatic colorectal, endometrial and gastric studies[48–52]. Results shown are in part based upon data generated by the TCGA Research Network (http://cancergenome.nih.gov/). Additional pancreatic data were obtained from[53,54]. Appendiceal exome information was derived from two studies in the literature[55,56]. Only high-level amplifications and homozygous deletions were considered for copy number changes. All mutations were included for the majority of genes, apart from those where more information is available regarding oncogenicity of specific mutation types such as *KRAS* (missense only) and *CDKN2A* (inactivating mutations only). The percentage of each genetic event in each tumor type was calculated and hierarchical clustering was performed in R using heatmap.2 defaults in gplot v 3.0.1[57].

**RNAseq**. Differential gene expression analysis was performed using Degust[58] with a false discovery rate threshold of $p < 0.05$. Significantly expressed genes were evaluated using STRING (v10.5)[59], and the heatmap created using normalized gene expression values and heatmap in R with default settings.

**Statistical analyses**. Statistical tests were performed in R (v3.3.0) as indicated in the main text and figure legends and in Supplementary Note 1. A two-tailed $p$-value of < 0.05 was considered as statistically significant except as otherwise indicated.

**Ethics statement**. MacCallum Cancer Centre Human/cThis study was approved by the Peter MacCallum Cancer Centre Human Research Ethics Committee, ID #14/76 and #01/38 and the Melbourne Health Human Research Ethics Committee #2011.248. Informed consent was obtained for all patients in the study. All relevant ethical regulations have been complied with.

**Reporting Summary**. Further information on research design is available in the Nature Research Reporting Summary linked to this article.

## Data availability

The exome and RNA sequencing data have been deposited in the European Genome-phenome database under the accession code EGAS00001003545. Other datasets referenced during the study are available from GSE39076 and as Supplementary Data in Ryland et al.[17] [https://doi.org/10.1186/s13073-015-0210-y]. All the other data supporting the findings of this study are available within the article and its supplementary information files (Source Data) and from the corresponding author upon reasonable request. A reporting summary for this article is available as a Supplementary Information file.

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

## Acknowledgements

This study was supported by the National Health and Medical Research Council of Australia (NHMRC) Grants #APP1045783 and #628434 and the Peter MacCallum Cancer Foundation. The authors acknowledge the Bioinformatics and Molecular Genomics Core Facilities of the Peter MacCallum Cancer Centre, which were supported by the Australian Cancer Research Foundation. We thank Sachintha Wijegunasekara and Ellen Gunn for technical assistance, and Margot Osinski (Royal Women's Hospital) for database assistance. The following cohorts were supported as follows: **CASCADE:** Supported by the Peter MacCallum Cancer Foundation. **AOCS:** The Australian Ovarian Cancer Study Group was supported by the U.S. Army Medical Research and Materiel Command under DAMD17-01-1-0729, The Cancer Council Victoria, Queensland Cancer Fund, The Cancer Council New South Wales, The Cancer Council South Australia, The Cancer Council Tasmania and The Cancer Foundation of Western Australia (Multi-State Applications 191, 211 and 182) and the National Health and Medical Research Council of Australia (NHMRC; ID400413 and ID400281). The Australian Ovarian Cancer Study gratefully acknowledges additional support from Ovarian Cancer Australia and the Peter MacCallum Foundation. The AOCS also acknowledges the cooperation of the participating institutions in Australia and acknowledges the contribution of the study nurses, research assistants and all clinical and scientific collaborators to the study. The complete AOCS Study Group can be found at www.aocstudy.org. We would like to thank all of the women who participated in these research programs. **COEUR:** This study uses resources provided by the Canadian Ovarian Cancer Research Consortium's - COEUR biobank funded by the Terry Fox Research Institute and managed and supervised by the Centre hospitalier de l'Université de Montréal (CRCHUM). The Consortium acknowledges contributions to its COEUR biobank from Institutions across Canada (for a full list see http://www.tfri.ca/en/research/translational-research/coeur/coeur_biobanks.aspx). **The Gynaecological Oncology Biobank at Westmead** is a member of the Australasian Biospecimen Network-Oncology group, which was funded by the National Health and Medical Research Council Enabling Grants ID 310670 & ID 628903 and the Cancer Institute NSW Grants ID 12/RIG/1-17 & 15/RIG/1-16. **OVCARE** receives core funding from The BC Cancer Foundation and the VGH and UBC Hospital Foundation. **Mayo Clinic**: National Institutes of Health (R01-CA122443, P30-CA15083, P50-CA136393); Mayo Foundation; Minnesota Ovarian Cancer Alliance; Fred C. and Katherine B. Andersen Foundation. **Edinburgh:** We extend our thanks to the Edinburgh Ovarian Cancer Database from which data were collected for this research and the Nicola Murray Foundation for supporting the Nicola Murray Centre for Ovarian Cancer Research.

## Author contributions

Apart from first and senior authors, all are listed in alphabetical order. P.E.A., M.Cr., S.B.F., C.B.G., M.K., J.P., K.R., and R.S. undertook pathology review of cases; G.A.-Y., K.A., A.B., D.D.L.B., M.B., Y.E.C., M.Ch., A.D.F., R.Du., N.F., C.G., N.F.H., J.H., D.G.H., C.J.K., K.R.K., S.H.K., C.L.P., A.M.M., J.N.M., O.M.M., L.M., D.M.P., J.P., A.N.S., G.S., G.C.T., and N.Tr. provided access to tissue and/or prepared tissue samples; G.A.-Y., K.A., M.S.A., S.A., M.B., M.Ch., R.Du,. N.F., S.F., A.M.H., G.Y.H., T.J., C.J.K., K.R.K., J.N.M., C.L.P., G.C.T., and N.Tr. provided clinical information; K.C.A., R.De., K.L.G., R.L., J.L., N.Th., M.J.W., Z.X., and M.Z. performed bioinformatics analyses; D.C., S.H., S.M.H., G.L.R., S.M.R., H.S., C.S., A.J.S., and M.C.T. performed nucleic acid extraction, PCR and Sanger sequencing. D.C., G.L.R., and T.S. performed sequencing library preparations. K.L.G. and G.L.R. coordinated the study. D.C., K.L.G., C.S., M.J.W., and M.C.T. analysed the data. Y.C.A., I.G.C., K.L.G., C.L.S., and M.J.W. conceived of and designed the study and were involved at all stages. K.L.G. prepared the figures and drafted the manuscript, which was then extensively edited by Y.C.A., D.D.L.B., D.C., I.G.C., A.D.F., M.K., S.H.K., J.N.M., C.L.S., C.G.F., and M.J.W. All other authors read and commented on the manuscript and approved the final version.

## Additional information

**Competing interests:** The authors declare no competing interests.



Dane Cheasley [1,26], Matthew J. Wakefield [2,3,26], Georgina L. Ryland [1,26], Prue E. Allan[1], Kathryn Alsop[1,3], Kaushalya C. Amarasinghe [1], Sumitra Ananda[1,4], Michael S. Anglesio [5], George Au-Yeung[1,3], Maret Böhm[6], David D.L. Bowtell[1,3], Alison Brand[7], Georgia Chenevix-Trench[8], Michael Christie[3,9], Yoke-Eng Chiew[7], Michael Churchman[10], Anna DeFazio [7], Renee Demeo[1], Rhiannon Dudley[11], Nicole Fairweather[11], Clare G. Fedele[1,3], Sian Fereday [1,3], Stephen B. Fox [1,3], C Blake Gilks[5], Charlie Gourley [10], Neville F. Hacker[12], Alison M. Hadley[13], Joy Hendley[1], Gwo-Yaw Ho [2], Siobhan Hughes[1], David G. Hunstman[5], Sally M. Hunter[1], Tom W. Jobling[14], Kimberly R. Kalli[15,28], Scott H. Kaufmann [15], Catherine J. Kennedy [7], Martin Köbel [16], Cecile Le Page [17], Jason Li[1], Richard Lupat[1], Orla M. McNally[3,18], Jessica N. McAlpine [5], Anne-Marie Mes-Masson[17,19], Linda Mileshkin[1], Diane M. Provencher[17,20], Jan Pyman[18,21], Kurosh Rahimi[17,20], Simone M. Rowley[1], Carolina Salazar[1], Goli Samimi[6], Hugo Saunders[1], Timothy Semple[1], Ragwha Sharma[7,22], Alice J. Sharpe[23], Andrew N. Stephens [11], Niko Thio[1], Michelle C. Torres [1], Nadia Traficante[1,3], Zhongyue Xing[1], Magnus Zethoven[1], Yoland C. Antill[24,25,27], Clare L. Scott [1,2,3,9,27], Ian G. Campbell [1,3,27] & Kylie L. Gorringe [1,3,27]

[1]Peter MacCallum Cancer Centre, Melbourne, Australia. [2]Walter and Eliza Hall Institute, Parkville, Australia. [3]The University of Melbourne, Melbourne, Australia. [4]Western Health, St. Albans, Australia. [5]University of British Columbia, Vancouver, Canada. [6]Kinghorn Cancer Centre and Garvan Institute of Medical Research, Darlinghurst, Australia. [7]Westmead Hospital, University of Sydney, Sydney, Australia. [8]Queensland Institute of Medical Research, Brisbane, Australia. [9]Royal Melbourne Hospital, Parkville, Australia. [10]Nicola Murray Centre for Ovarian Cancer Research, Cancer Research UK Edinburgh Centre, University of Edinburgh, Edinburgh, UK. [11]Hudson Institute of Medical Research, Clayton, Australia. [12]The University of New South Wales, Sydney, Australia. [13]Royal Brisbane and Womens Hospital, Brisbane, Australia. [14]Monash Medical Centre, Clayton, Australia. [15]Mayo Clinic, Rochester, MN, USA. [16]The University of Calgary, Calgary, Canada. [17]CRCHUM, Montreal, Canada. [18]Royal Womens Hospital, Parkville, Australia. [19]University of Montreal, Montreal, Canada. [20]Centre Hospitalier de L'Université de Montreal, Montreal, Canada. [21]Royal Children's Hospital, Flemington, Australia. [22]NSW Health Pathology, Sydney, Australia. [23]Monash University, Clayton, Australia. [24]Cabrini Health, Malvern, Australia. [25]Frankston Hospital, Frankston, Australia. [26]These authors contributed equally: Dane Cheasley, Matthew J. Wakefield, Georgina L. Ryland. [27]These authors jointly supervised: Yoland C. Antill, Clare L. Scott, Ian G. Campbell, Kylie L. Gorringe. [28]Deceased: Kimberly R. Kalli.

