## [Peer Review File · Nature Communications]

Reviewers' comments:

Reviewer #1 (Remarks to the Author):

The paper contains a lot of information but it is very difficult to follow and the language is both idiosyncratic and esoteric

there are too many acronyms, neologisms (truncal events) and poor wording (ovarian origin).

The authors should go over carefully the logic behind the model and provided the data in a simplified way that makes it accessible to a larger readership

The material on extra-ovarian origin of these cancers is marginal and should be summarized in two or three sentences in the discussion not as a lead. They are obviously not metastatic pancreatic cancers no need to dwell on it.

provide a few sentences of background on the genetic data applied to tumor evolution. what do we expect to see.

I am not sure who drafted this version but I suggest that one of the other authors should give it a try.

Reviewer #2 (Remarks to the Author):

This is a very solid MS from an excellent group studying an important and difficult clinical question—what is the genomic landscape of primary ovarian mucinous carcinoma. This paper is an important landmark study for this disease.

Strengths of the MS are:

1. The large sample collection with detailed pathological review. Note that this is probably the largest survey to date with genomic profiling and the pathology input is critical for meaningful comparisons
2. High quality sequencing data and interpretation
3. The warm autopsy case is illustrative both genomically and highlighting poor outcomes for these patients

Weakness are:

1. The work is mostly confirmatory of existing smaller series
2. Given the strong RAS drivers, are there other more novel observations that can be made about transformation, particularly in the warm autopsy case? Was there whole genome duplication in the metastatic disease in this case? Can the authors contrast their data more strongly with existing hypotheses about RAS-driven transformation (aberrant G2, mitotic checkpoint controls and missegregation)? Did the frequency of fold-back inversions increase?

James Brenton

Response to Reviewers' comments:

Reviewer #1 (Remarks to the Author):

“The paper contains a lot of information but it is very difficult to follow and the language is both idiosyncratic and esoteric

there are too many acronyms, neologisms (truncal events) and poor wording (ovarian origin).

The authors should go over carefully the logic behind the model and provided the data in a simplified way that makes it accessible to a larger readership”

...

“I am not sure who drafted this version but I suggest that one of the other authors should give it a try.”

We have consulted our institute’s Research Communications Officer (Dr Clare Fedele, who is herself a published scientist in a different field) for readability by a non-specialist. She has revised the manuscript for clarity, simplified/explained some terms that a non-expert may not be familiar with, and removed some acronyms (for example “EOM”, “G1”, “G2”, “G3”). We believe this has improved the appeal of the work to a broader audience.

“The material on extra-ovarian origin of these cancers is marginal and should be summarized in two or three sentences in the discussion not as a lead. They are obviously not metastatic pancreatic cancers no need to dwell on it.”

We respectfully disagree with the reviewer on this point. The origin of these cancers is extremely controversial in the field (e.g. Table 1 in Karnezis et al. Nature Reviews Cancer 2017 17:65–74) and of high relevance for their clinical management (e.g. “An accurate diagnosis of primary mEOC or metastatic disease is mandatory for different therapeutic approaches” Ricci et al. Int J Mol Sci. 2018 19(6): 1569). We also believe that given the genetic similarity of MOC to pancreatic adenocarcinoma, and other work proposing a shared origin (e.g. Elias et al. J Pathol. 2018 246(4):459-469), that this specific point should be clearly made.

“provide a few sentences of background on the genetic data applied to tumor evolution. what do we expect to see.”

We have added the several sentences to the introduction to clarify the background and hypothesised outcomes.

Reviewer #2 (Remarks to the Author):

This is a very solid MS from an excellent group studying an important and difficult clinical question—what is the genomic landscape of primary ovarian mucinous carcinoma. This paper is an important landmark study for this disease.

Strengths of the MS are:

1. The large sample collection with detailed pathological review. Note that this is probably the largest survey to date with genomic profiling and the pathology input is critical for meaningful comparisons
2. High quality sequencing data and interpretation
3. The warm autopsy case is illustrative both genomically and highlighting poor outcomes for these patients

Weakness are:

1. The work is mostly confirmatory of existing smaller series

We agree our results are consistent with previous smaller studies, but point to the finding of an association of genomic aberration load with outcome as one example of a novel (and potentially translatable) observation that is only possible with our large sample size. In addition, previous series contain few, if any, true high-grade MOC, and these have not previously been shown to be related to borderline precursors.

2. Given the strong RAS drivers, are there other more novel observations that can be made about transformation, particularly in the warm autopsy case? Was there whole genome duplication in the metastatic disease in this case? Can the authors contrast their data more strongly with existing hypotheses about RAS-driven transformation (aberrant G2, mitotic checkpoint controls and missegregation)? Did the frequency of fold-back inversions increase?

The metastatic samples did have genome duplication (added to results section page 11). Fold-back inversions did increase in the metastases compared to the primary tumour from 4-6 (7-9%) to 16-20 (12-13%) in the metastases. This information has been added to the main manuscript page 11 and is present in the Supplementary Note (under the "structural analysis" section, where the actual numbers have been added to the percentages previously provided). We also have discussed the role of KRAS and TP53 further in the Discussion (page 13).

Reviewers' comments:

Reviewer #1 (Remarks to the Author):

The paper improved from the last version

The paper contains interesting information and is potentially important and would be an excellent paper if it were well- written.

I fail to see that it is real question whether or not mucinous ovarian cancer are actually metastatic pancreatic cancer when the former have a 80% five year survival and the latter have a 5% five year survival. They are obviously not the same thing at a group level - the question is can the pathologist distinguish between the two for a single patients at diagnosis based on histology? and does the distinction impact on treatment? are the molecular and genetic markers helpful in this regard? In page 6 they say that the defined 249 primary MOC from 500 potential mucinous ovarian cancers. how did they do this? Is this not putting the cart before the horse? How do they know they were not pancreatic cancers? So it is not hard to tell them apart

Reviewer #2 (Remarks to the Author):

The manuscript has been further improved with editing and new information about whole genome duplication for the warm autopsy case.

All points have been fully addressed.

James D. Brenton

Response to reviewers

Reviewer #1 (Remarks to the Author):

The paper improved from the last version

The paper contains interesting information and is potentially important and would be an excellent paper if it were well- written.

I fail to see that it is real question whether or not mucinous ovarian cancer are actually metastatic pancreatic cancer when the former have a 80% five year survival and the latter have a 5% five year survival. They are obviously not the same thing at a group level - the question is can the pathologist distinguish between the two for a single patients at diagnosis based on histology? and does the distinction impact on treatment? are the molecular and genetic markers helpful in this regard? In page 6 they say that the defined 249 primary MOC from 500 potential mucinous ovarian cancers. how did they do this? Is this not putting the cart before the horse? How do they know they were not pancreatic cancers? So it is not hard to tell them apart

A long list of criteria has been established to distinguish primary ovarian from metastatic mucinous carcinomas including gross features (laterality, size) and microscopic features (surface involvement, nodularity, signet ring cells, among several others) [Refs Lee & Young, Am J Surg Pathol. 27(3):281-92 2003, PMID: 1260488; Stewart et al., Int J Gynecol Pathol. 33(1):1-10, 2014, PMID: 24300528]. Many primary ovarian mucinous carcinomas can now be confidently classified as primary by an experienced pathologist. The most commonly encountered metastases in this scenario are not pancreatic, but lower gastrointestinal, followed by gastric and breast carcinomas. However, distinguishing primary ovarian mucinous carcinoma from metastatic upper gastrointestinal primary (including pancreas) is

one of the last challenging resorts in gynecopathology (Ackroyd et al., Gynecol Oncol Rep. 28:109-115, 2019, PMID: 30997376). Because none of the criteria even in combination are entirely specific, in some cases, a vague diagnosis of mucinous adenocarcinoma favouring ovarian primary/ favouring metastatic is rendered, with the recommendation to rule out metastatic disease by clinical investigations such as imaging for pancreas or endoscopy for stomach. This means that **at the time of diagnosis** it can be challenging to assess the primary site for a subset of mucinous adenocarcinoma and there are no ancillary molecular or genetic tests in widespread clinical use. This decision does indeed impact treatment, i.e. a stage I primary ovarian mucinous carcinoma may not receive further treatment after surgery (given the good outcome) while a metastatic pancreatic adenocarcinoma should receive maximal treatment. The choice of chemotherapy is also strongly influenced by tissue of origin in most current guidelines. Thus, the determination of primary status is still an important unmet need in clinical practice, as recognised in the GCIg consensus document (Leary et al., Annals of Oncology 28: 718–726, 2017, PMID:27993794).

In our study, we reviewed over 500 potential primary ovarian mucinous tumours using an integrated approach combining multiple pathological criteria and the clinical outcome. These cases were diagnosed over a long time period with a less stringent application of above criteria. Therefore, we suspected a potential influx of metastatic adenocarcinomas. Our expert pathologists, all of whom had many years of experience, applied contemporary criteria, and cases interpreted as likely metastatic were removed from the study. We acknowledge that we may have introduced a bias, however, we believe that the larger danger would have been to include potential metastatic cases in the study. Our sample size still comprises the largest number of mucinous ovarian carcinomas studied to date, and the molecular differences we observe could be useful in the future for determining origin.

We clarified the controversy and relevance of determining the primary origin of MOC in the introduction (first paragraph):

“Accurate diagnosis of primary MOC remains challenging, with a metastatic tumor from the lower gastrointestinal tract the most common alternative⁴. Knowing primary *versus* metastatic status strongly influences therapy selection, since most international guidelines indicate that first line therapy should be based on the tissue of origin^{5, 6, 7}. Ovarian platinum-based therapies have low response rates for MOC⁸ and because of the morphological similarities with colorectal mucinous tumors, a colorectal treatment regimen was proposed⁹. The difficulties in diagnosis alternatively led to the suggestion that all mucinous tumors could be treated with similar targeted therapies, regardless of origin^{10, 11}. However, both approaches assume molecular similarities across mucinous tumors, which is currently unknown.”

Reviewer #2 (Remarks to the Author):

The manuscript has been further improved with editing and new information about whole genome duplication for the warm autopsycase.

All points have been fully addressed.

REVIEWERS' COMMENTS:

Reviewer #1 (Remarks to the Author):

The concerns have been addressed

Reviewer #2 (Remarks to the Author):

No further comments. The REMARK diagram and revised paragraph are very helpful in demonstrating the quality of the work and the criteria for inclusion.

James D. Brenton